# A Study on the Gaze Range Calculation Method During an Actual Car Driving Using Eyeball Angle and Head Angle Information

**DOI:** 10.3390/s19214774

**Published:** 2019-11-02

**Authors:** Keiko Sakurai, Hiroki Tamura

**Affiliations:** Faculty of Engineering, University of Miyazaki, 1-1, Gakuen Kibanadai-nishi, Miyazaki-shi 889-2192, Japan; sakurai.keiko.u6@cc.miyazaki-u.ac.jp

**Keywords:** gaze range calculation method, gaze estimation, head angle, eye tracking device, template matching, gaze range

## Abstract

Car operation requires advanced brain function. Currently, evaluation of the motor vehicle driving ability of people with higher brain dysfunction is medically unknown and there are few evaluation criteria. The increase in accidents by elderly drivers is a social problem in Japan, and a method to evaluate whether elderly people can drive a car is needed. Under these circumstances, a system to evaluate brain dysfunction and driving ability of elderly people is needed. Gaze estimation research is a rapidly developing field. In this paper, we propose the gaze calculation method by eye and head angles. We used the eye tracking device (TalkEyeLite) made by Takei Scientific Instruments Cooperation. For our image processing technique, we estimated the head angle using the template matching method. By using the eye tracking device and the head angle estimate, we built a system that can be used during actual on-road car operation. In order to evaluate our proposed method, we tested the system on Japanese drivers during on-road driving evaluations at a driving school. The subjects were one instructor of the car driving school and eight general drivers (three 40–50 years old and five people over 60 years old). We compared the gaze range of the eight general subjects and the instructor. As a result, we confirmed that one male in his 40s and one elderly driver had narrower gaze ranges.

## 1. Introduction

In recent years, car accidents have occurred due to elderly people and patients with higher brain dysfunction. Developing a driving ability evaluation system for the elderly and patients with higher brain dysfunction is of great necessity. However, establishing criteria for judging the driving ability of these groups is difficult [1]. The methods for evaluating the driving ability of patients with higher brain dysfunction include off-road assessment by neuropsychological examination and on-road evaluation in an actual vehicle. Of these evaluations, the most effective is on-road evaluation in an actual vehicle because it most closely resembles real world scenarios [2]. Therefore, in order to systematically advance the road driving evaluation of elderly drivers and patients with higher brain dysfunction, it will be necessary that medical institutions and driving schools work together to make a new evaluation system.

About 90% of the information obtained when driving a car is said to be visual information [3]. Owsley et al. [4] modeled the relationship between eye health, effective field of view, and car accidents involving elderly people (mean age = 70 years old). The authors found that the effective field of view is most related to car accidents involving elderly people [4]. Therefore, the effective field of view range is greatly involved in the driving ability of motor vehicles.

While driving, people often move their gaze extensively, and in most cases the head and eye positions move simultaneously [5,6,7,8]. This relationship between head and eye movement is called eye–head cooperative movements, and human beings perform this movement reflexively, so the analysis of the movement of the head and eyes has been of great interest in recent years [9]. Therefore, we previously proposed the head angle detection method using an RGB-D (for Red, Green, Blue plus Depth) sensor [10]. RGB-D sensor is a specific type of depth sensing devices that work in association with an RGB camera. RGB-D sensor can augment the conventional image with depth information (related with the distance to the sensor) in a per-pixel basis. In addition, we proposed a system that combined RGB-D sensor and an electrooculogram (EOG) sensor to estimate the gaze [11,12]. However, this proposed system did not have high enough accuracy to use for car driving evaluation (average error of eyeball angle is over 10 degrees). Furthermore, the existing systems require the placement of RGB-D sensors, but this sensor is not suitable for use in tight spaces, such as inside a car because it requires about 1 m distance between the subject and the sensor. 

Considering the problems of the previously proposed system, this paper proposes a method that enables high-performance line-of-sight estimation with a single device and constructs a line-of-sight estimation system at low cost. In this paper, eye movement is acquired using an eye tracking device (TalkEyeLite [13]), and the head angle is estimated by image processing technology using the TalkEyeLite’s view camera. We focused on the gaze range while driving. In our proposed image processing technology, template matching was performed using the view camera attached to TalkEyeLite. Our proposed method requires only TalkEyeLite, so, it can be said that our proposed method is suitable to use in the limited space available inside a typical Japanese car. To test our proposed system, we calculated the gaze range of nine subjects (one driving school instructor, 47 years old) and eight general men subjects, of whom three were men, 40–50 years old and five were over 60 years old). Furthermore, we calculated the gaze range of nine subjects and considered the car driving ability using the gaze range. 

## 2. Previous Study about Gaze Estimation Method for Driving the Car

With the increase in car driving accidents, studies of detection of fatigue during driving using speech recognition [14] and detection of sleepiness using percentage of eyelid closure (PERCLOS) [15] have been conducted. Although fatigue and sleepiness can be detected using these methods, they are insufficient as a method for determining driving ability.

Several studies have been done on gaze estimation during driving [16,17]. Martin [16] modeled drivers’ gaze behaviors in order to predict lane changing and lane keeping behaviors. However, this research was limited to simulation, and the pragmatic application to actual vehicles was not conducted.

A driver’s gaze can also be estimated from their head movement. Gaze estimation has been performed by tracking the features of the driver’s head using a monocular camera attached to the steering column [18]. Although there are some studies [18,19,20,21,22] that estimated the gaze direction via head movement, the problem with this method is that the accuracy is not high enough to be reliable for driving ability evaluation. In order to improve the gaze measurement accuracy, there has been research on gaze estimation using the movement of both the head and the eyes—a system that can be used while driving a car [23,24]. This research has utilized an “omnicam (3D panoramic camera)” and a “noninvasive stereo camera” [23,24]. Although these two methods can be used anywhere, they require recognition of the head and eyes via image processing, and the recognition rate is not enough. 

In the case of car driving evaluation, it is necessary to determine whether the driver is looking at a pedestrian or a traffic light. However, in previous research [5,16,17,18,19,20,21,22,23,24], it was not possible to know whether the driver is aware of the information on the road (e.g., pedestrians, traffic lights, road signs). Our proposed method can finally determine what specifically the driver is looking at using the image from the view camera. Therefore, our system is a usable method for calculating the gaze range while driving.

## 3. Measurement System

### 3.1. Measurement System Using TalkEyeLite

The calculation of the eye movement angle of TalkEyeLite is a pupil image processing method. TalkEyeLite [13] is a wearable eye movement measurement system that connects directly to the processing computer and uses a USB camera for eye detection and recording visual field images. Through its USB camera, TalkEyeLite can track the subject’s pupils, enabling us to record what the subject is seeing and focusing on. The overlay display on the visual field image includes the center axis of the gaze of both eyes in addition to the left and right viewpoints. 

Figure 1a is a picture of TalkEyeLite goggles with labels indicating the view camera and the eye camera, and Figure 1b is a picture of a lab member wearing TalkEyeLite goggles.

The gaze point of both eyes is indicated by the blue cross points in Figure 2. The eye motion analysis program can analyze visual videos using the measurement data recorded by TalkEyeLite, enabling us to know which target the subject was looking at. The angle of convergence can be calculated using the angle data from both eyes. The angle of convergence is expressed in degrees and the value ranges from −180 degrees to +180 degrees.

### 3.2. The Head Angle Estimation Method Using Template Matching

We proposed a head angle estimation method using template matching. We used the viewing camera of TalkEyeLite to obtain the head movement information. Although the head angle estimation method using the RGB-D sensor needs to set a sensor, the head angle estimation method using the proposed template matching uses a field of view camera attached to TalkEyeLite. Therefore, our proposed method has the merit that it can be easily used in various environments. Since the estimation of the line of sight and the head angle is performed during actual car driving, it is necessary to use a method that can estimate the line of sight given the restricted physical space available for equipment and measurement; template matching accomplishes this. We installed three templates as shown in Figure 3 (Marker1, an upward triangle; Marker2, a circle; and Marker3, a downward triangle) on the front glass of the car. We performed template matching using these three template images to estimate the head angle as described by Figure 4.

This system required calibration prior to data collection onset. We defined the forward-facing position of the head as angle 0 (deg). We placed the markers (Marker1, Marker2, and Marker3) on the dashboard of the car, with one of the three markers directly in front of the driver (0 (deg)). We asked the driver to face forward for five seconds in order to record the reference position for each of the three markers. These reference positions were used for template matching. The driver’s head angle was calculated after determining the head movement amount “A” using the difference between the current marker’s position and the reference position. Due to movement on the roll axis (Figure 5), it was possible for two markers to be found during image processing; this problem is addressed in the next section. 

#### Problems Caused by Inclination of the Head

As for the inclination of the head, there are three axis directions: the yaw axis, the pitch axis, and the roll axis (Figure 5). The yaw angle and the pitch angle are necessary for gaze estimation. The yaw angle and the pitch angle of the head are calculated using differences in the recorded marker coordinates. 

However, if the yaw axis and the pitch axis change while the roll axis of the head is inclined, the head orientation angle cannot be estimated correctly. An example is shown in Figure 6. When the head roll axis (θ in Figure 6) is tilted and if the b (deg) moves to the right, the marker position moves a pixel. Therefore, when the roll axis rotates, we must first determine the roll angle. After calculating the roll angle, coordinate transformation was performed. The deviation of the estimated angle due to the movement of the roll angle is shown in Figure 6. The deviation of the yaw axis can be calculated by a-acosθ and the deviation of the pitch axis by asinθ. By considering these deviations, we obtained coordinates converted to zero roll angle, solving the problem caused by inclination of the head. 

The closer the coordinates are to the center of the camera, the smaller the amount of head movement. Therefore, the yaw angle is obtained from the marker close to the center of the x coordinate, and the pitch angle is obtained from the marker close to the center of the y coordinate. The roll angle calculation formula (Equation (1)) and the yaw angle (Equation (4)) and pitch angle (Equation (5)) calculation formulas in consideration of the roll angle are given below.

• Calculation of the roll angle

The formula for calculating the roll angle is shown in Equation (1). Letting the coordinates of the two found markers be ($x_{1}$, $y_{1}$) and ($x_{2}$, $y_{2}$), the roll angle can be calculated by the Equation (1), assuming that $x_{2}$ > $x_{1}$.
(1)$$\theta =\tan^{-1}\left(\frac{y_{2}-y_{1}}{x_{2}-x_{1}}\right);$$
θ: roll angle, ($x_{1}$, $y_{1}$) ($x_{2}$, $y_{2}):\;\mathrm{the}\;\mathrm{coordinates}\;\mathrm{of}\;\mathrm{the}\;\mathrm{two}\;\mathrm{markers}\;\mathrm{found}$.

• Calculation of yaw angle and pitch angle

The yaw angle and the pitch angle were calculated using the movement amount for a pixel of the marker. The variables *x* and *y* indicate the coordinates of the marker before the coordinate transformation, whereas the variables *x’* and *y’* indicate the coordinates after transformation. Equations (2) and (3) are the equations for rotation conversion. Equations (4) and (5) are the equations for calculating yaw angle and pitch angle. For example, if the coordinate of the marker moved a pixel horizontally, the head inclined at an angle shown in Equation (4).
(2)$$x^{\prime }=x\cos\theta -y\sin\theta$$
(3)$$y^{\prime }=x\sin\theta +\mathrm{ycos}\theta$$
(4)$$Yaw\;face\;angle=\frac{a}{\frac{Camera\;resolution}{Camera\;viewing\;angle}}\;\left(deg\right)$$
*a:* horizontal movement amount of marker (pixel)
(5)$$Pitch\;face\;angle=\frac{b}{\frac{Camera\;resolution}{Camera\;viewing\;angle}}\;\left(deg\right)$$
*b:* vertical movement amount of marker (pixel)

## 4. Improved Measurement Method 

The measurement system described above in Section 3 has the following two problems:
The gaze estimation angle range is narrow (about 42 degrees).The processing time is long.

In Section 4, we describe our proposed solutions to these problems.

### 4.1. Expansion of the Gaze Estimated Range

In previous methods, there was a problem with the narrow range of the estimated head angle. Therefore, we expanded the estimable range by including a new marker, and we evaluated the accuracy of the improved method. In order to expand the estimation range, a new star marker (Marker4 in Figure 7) was added to the left end. In order to calculate head movement and angle information, it is necessary to record a reference position for each marker. As described previously, subjects performed calibration by facing forward for five seconds before beginning to drive. The range of calibration is shown in Figure 8. Calibration is performed with the forward-facing position. Marker4 is excluded from calibration because Marker4 does not appear on the screen when facing the front. Therefore, the reference position of the star marker was calculated using the ratio of α and β as shown in Figure 8. The coordinates of Cx were obtained by transforming Equation (6), as in Equation (7), to Equation (9). Using the obtained Cx, the accuracy of the head orientation calculation was estimated (Figure 9).
(6)$$\mathrm{Bx}=\;\frac{\beta \mathrm{Ax}+\alpha \mathrm{Cx}}{\alpha +\beta }$$
(7)(α+β)Bx=βAx+αCx
(8)−αCx=−αBx−βBx+βAx
(9)$$\mathrm{Cx}=-\frac{1}{\alpha }\left\{\;\beta \left(\mathrm{Ax}-\mathrm{Bx}\right)-\alpha \mathrm{Bx}\right\}$$
(Cx < Bx < Ax)

### 4.2. The Method of Reducing Processing Time

In our system, OpenCV was used for template matching. Since our proposed method searched template matching over the entire image range, it took time to process. Therefore, in this section, we propose an updated method to shorten the search time by narrowing the search range. We also verified whether highly accurate template matching can be performed with this method. 

#### 4.2.1. Positional Relation of Markers to Reduce Search Range

First, we described the positional relationship of the markers used for narrowing the search range. The markers are searched in the following order: the upward triangle marker (Marker1), then the circle marker (Marker2), then the downward triangle marker (Marker3), and finally the star shape marker (Marker4). For each marker, the search range was decided using the positional relation between previously searched-for markers as applicable. The search range of each marker was determined by the positional relationships between the markers, as follows:
Marker2 is more to the right than Marker1.Marker3 is more to the right than Marker1 and Marker2.Marker4 does not appear in the same frame as Marker1 and Marker2.Marker4 is more to the right than Marker3.

Considering these four positional relationships, we reduced the search range as illustrated in Figure 10, Figure 11 and Figure 12. Only the yellow highlighted area was included in the search range for each respective marker.

#### 4.2.2. Dividing the Search Range

Considering the method of Section 4.2.1, we next introduced a method for reducing the processing time by: (1) dividing the search range into sections and (2) performing template matching beginning from the section where there was a high probability that markers are present. 

Only rarely does a driver’s head posture vary greatly up and down. Therefore, it was unlikely for a marker to exist above and below the search range. To speed up processing, it’s important to perform template matching beginning from the range where there the highest possibility that a marker is present. Therefore, we divided the search range into three sections and searched for markers within each section.

When we divide the search range into three sections, a marker may span two ranges as in Figure 13. In this case, it was more likely for template matching to fail in both ranges, causing the drop of recognition rate and the extension of processing time. To prevent these problems, we expanded the search range up and down by ((length of the template image) – 1) (pixel). Figure 14 illustrates the extended search range.

Template matching was performed in the following order: first, central section; second, upper section; and third, bottom section. If the degrees of similarity in a section exceeded a constant value, template matching was performed in the section.

#### 4.2.3. Extending the Search Range

We extended the search range to increase the recognition rate from template matching. In the proposed method from Section 3.2, if the marker was found in a previous frame, template matching was performed around the position of the marker in the previous frame. However, the template matching would fail if the driver moved their head a large distance before turning the car left or right because the marker would move out of the search range. Additionally, the division of the search range into sections lengthened the processing time due to failed template matching. To address these issues, we enabled the system to alter the search range based on the quantity of movement between the current and previous frames. For example, if the marker moved significantly right from two frames before to one frame before, the search range was extended to the right side by a constant range. Figure 15 illustrates an example. We judge that the marker moved significantly from the head movement angle and the eye movement angle. We also judged from the visual field camera image.

## 5. Verification of the Accuracy of the Proposed Method

First, in order to verify the accuracy of the proposed method, verification experiments were performed indoors in an ideal environment. This was followed by a verification experiment using a trial run at the driving school. 

### 5.1. Indoor Experiments: Conditions and the Results

In order to grasp the error of the head angle correctly, the indoor experiment was carried out assuming the optimum environment. The indoors experimental environment is shown in Figure 16. The mark (⌧) indicates 0 to 60 degrees and is not a marker for template matching. Markers were set at 0, 10, 20, 30, 40, 50, and 60 degrees; 0 degrees is directly in front of the subject. The subject was instructed to view the markers from 0 degrees to 60 degrees in the following order: 0 degrees, 10 degrees, 20 degrees, 30 degrees, 40 degrees, 50 degrees, and 60 degrees. The subject was one woman, and the experiment was carried out once. Each marker had a true measured value as well as a value calculated using our gaze estimation method. An error was calculated based on the gaze estimation result. The result is shown in Table 1. The average error was 4.1 degrees. The recognition accuracy of template matching is shown Table 2.

A list of representative studies of head angle estimation method is shown in Table 3. 

Table 3 shows that our proposed method has the lowest mean error compared to the other methods.

We also consider the gaze estimation method. Since the eye movement angle and head angle work in coordination, the eye gaze angle is the sum of the estimation results of the eye movement angle and the head estimation angle from TalkEyeLite. So, we represented the gaze angle *E*(*t*) by *f*_1_(*t*) and *f*_2_(*t*) shown in Equation (10).
*E*(*t*) = *f*_1_(*t*) + *f*_2_(*t*)(10)
*f*_1_(*t*): the eye movement angle;*f*_2_*(t)*: the head angle;*t*: the number of sampling data.

So far, we performed gaze estimation using three gaze estimation methods.

1. EOG and RGB-D sensor 

EOG was used for gaze estimation. In the eye movements, a potential across the cornea and retina exists, and it is a source of EOG. We chose the RGB-D sensor to determine the direction of the head, and we proposed the system that estimates both the direction at the same time using the RGB-D sensor and the EOG sensor and estimated the gaze high accuracy. However, we need to attach the electrodes to use EOG. Also, the average error of the eyeball angle is over 10 degrees [30].

2. Tobii

Tobii is an eye tracking system using a camera [31]. Tobii can use the gaze to interact with computers and machines without mounting. Tobii provides a very efficient eye-tracking (accuracy <0.6°) naturally. However, Tobii was not possible to pursue the eyeballs’ position in the angle of 36° or more. Moreover, it is not suitable for wide-range gaze estimation because the face angle cannot be estimated.

3. Proposed Method

Our proposed method using TalkEyeLite can measure both head movement and eye movement. The eye angle is calculated by TalkEyeLite. When the eyeball is irradiated with weak infrared light, a reflected image of the light source is generated on the refractive surface of the cornea or crystal. This reflection image, called the Purkinje–Sanson image, has the characteristic of little movement relative to the pupil during eye movement. Both are photographed with a camera, and the “eye movement angle” is calculated from the positional relationship. TalkEyeLite provides an efficient eye-tracking (accuracy <3.0°, according to our results). The proposed method is suitable for wide-range gaze estimation because we set freely position markers. However, since TalkEyeLite is a goggle type device, the proposed method has a load to wear. 

The method using EOG is inferior to both TalkEyeLite and Tobii in terms of accuracy. Tobii has high precision, but Tobii cannot capture eye position in a wider range than 36°. Therefore, the third proposed method using TalkEyeLite and four markers has the advantageous ability to obtain both the eye angle and the head angel in wide range over 36°.

### 5.2. Outdoor Experiments (in an Actual Car): Conditions and the Result

We evaluated the accuracy of template matching and the shortened processing time during on-road car driving. The subject was a driving school instructor who drove the driving school course four times. 

Table 4 shows the marker recognition performance evaluation results. Table 5 shows the processing time for template matching after adding the search range expansion function and the processing time reduction function. By comparing three representative videos before and after adding the functions, we found that the time could be shortened considerably in each case. We also confirmed that the template matching ratio did not decrease, and the time could be shortened without degrading the performance. 

## 6. Proposed Driving Evaluation Method

In this section, we introduce the evaluation method of this system. 

• System Overview

The output screen of this system is shown in Figure 17. Our system showed the head angle, the gaze angle, the driver’s gaze (red circle), and a classification of the area that the driver was looking at. There were nine area classifications: “front,” “rearview mirror,” “meter,” “right mirror,” “right,” “right back,” “left mirror,” “left,” and “left back.” 

### 6.1. The Performance of Gaze Estimation (During Actual Car Driving)

In order to verify the gaze estimation accuracy of our system in a real car driving environment, we conducted the following experiments. To run the test course once required approximately eight minutes. From the measurement data, we randomly extracted 30 points around five seconds before and after left turns, right turns, and continued straight driving. For each of the 30 points, we determined whether the judgment area of the system matched the field of view of the actual driver. From this, we calculated the recognition rate. Table 6 shows the results of an outdoor experiment with one general driver and the “Judgment Flag” column of Table 6 indicates at which area our system judged the driver to be looking. The correct judgment rate was 93%, and the correct result was obtained in 28 out of 30 situations. For the two situations with judgment error (number 10 and 12), one error was caused by incorrect template matching, and the other error was caused by incorrect eye measurement.

### 6.2. Evaluation of the Gaze Range

In this study, gaze estimation was performed to evaluate the gaze range of elderly people during driving. The gaze range means a person’s ability. The eye movement angle was obtained by TalkEyeLite shown in Chapter 3, and the head movement angle was calculated using the template matching method. The gaze range was estimated using the value obtained by adding the head angle and the eyeball angle as the gaze. In order to evaluate participants’ gaze ranges using this system, on-road car driving experiments were conducted at the driving school.

To evaluate the performance of this system in evaluating gaze range, an experiment was conducted using an actual motor vehicle at the driving school. The subjects were one driving school instructor and eight general subjects (subject A: 60–69 years old, subject B: 60–69 years old, subject C: 60–69 years old, subject D: 60–69 years old, subject E: 70–79 years old, subject F: 40–49 years old, subject G: 40–49 years old, and subject H: 40–49 years old). The driving school instructor drove the test course four times, whereas each of the eight general subjects drove the course once. The data of the driving school instructor was taken as teacher data. 

In order to compare the gaze ranges of the elderly participants and the instructor, the head estimation angle, eye movement angle, and gaze angle were compared using histograms. Figure 18 is an example histogram of a participant’s gaze. The vertical axis represents the number of occurrences, the horizontal axis represents the X direction (left and right for the participant; the positive direction on the axis is right, the negative direction on the axis is left), and the depth represents the Y direction (up and down for the participant; the positive direction on the axis is up, the negative direction on the axis is down). A gaze range calculated with Gaussian fitting (95% confidence interval) is given by the shaded area. In other words, a wide shaded area indicated that the participant’s gaze range during driving is wide. 

Figure 19 and Figure 20 are histograms of the head angle, the eye movement, and the gaze during car driving, which means the head angle, eye movement angle, and gaze angle are distributed in the area during running. Figure 18 shows the histograms from the driving school instructor, and Figure 19 shows the histograms from subject B (60–69 years old). The amount of data varied by subject.

When the threshold value is 5%, the maximum value in the X direction is X_max_ and the minimum value is X_min_. We defined X as X_max_–X_min_. Similarly, Y is Y_max_–Y_min_ in the Y direction. The gaze range was calculated by X × Y, and the gaze ranges of the instructor and the general subjects (subject A to subject H) were compared (Figure 21). We found that subject B (60–69 years old) and subject G (40–49 years old) had narrow gaze ranges.

## 7. Conclusions

In this paper, we calculated the gaze range of nine subjects and considered the car driving ability using the gaze range. Since gaze range greatly contributed to car accidents, we evaluated driving ability by focusing on gaze range in this paper. We mainly carried out the following three things:(1)In order to extend the range in which head angle estimation can be performed, we proposed the head angle estimate method using markers, and we evaluated the accuracy of this method;(2)Shortening the processing time of template matching, we changed the range that performed template matching;(3)The gaze range during driving was compared between eight general subjects and one driving school instructor. As a result, we found that the gaze ranges of two subjects were extremely narrow.

We expanded the range of head angle estimation using the template matching method. The average error of estimated head angle was 4.1 degrees, and the recognition accuracy of template matching was 90.3%, which were enough for gaze estimation.

In order to evaluate the shortening result of the processing time of template matching, the accuracy under the situation in the vehicle was evaluated. As a result of verifying the three videos, we found that the time could be shortened considerably in each case. Processing time was been reduced by 28%.

The gaze range while driving plays a major role in driving a car. In particular, elderly people generally have a narrow field of view [32,33], but there has not been much research on the gaze range during actual vehicle driving. In this paper, we compared the gaze range with one instructor and eight general subjects and considered driving ability. In the evaluation, we found that subject B (60s) and subject G (40s) had a narrower gaze range than the instructor. By performing Gaussian fitting, we evaluated the driving ability based on the gaze range by comparing the gaze range of the instructor and the general subjects. 

The future methods for evaluating driving ability may benefit from considering both fields of view range and other aspects of driving ability (e.g., attention to road signs and lights). Errors in template matching are caused by the change in the shape of the template when the driver’s head moves a large distance or at a high speed. In the future, we will introduce machine learning techniques to address this issue in an attempt to reduce errors. 

TalkEyeLite requires wearing goggles, and the weight of the goggles is a bit on the user load. In the future, there is the possibility of developing an evaluation using devices with less load.

## Figures and Tables

**Figure 1 sensors-19-04774-f001:**
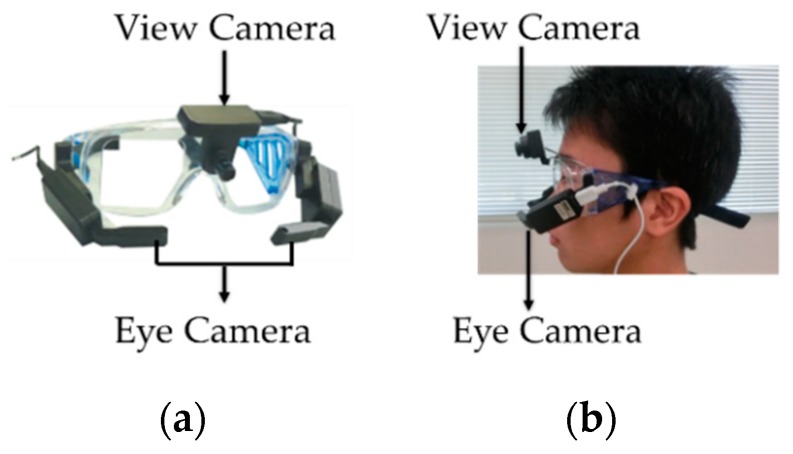
The measurement system of TalkEyeLite. (**a**) TalkEyeLite Goggles. (**b**) Lab member wearing TalkEyeLite goggles.

**Figure 2 sensors-19-04774-f002:**
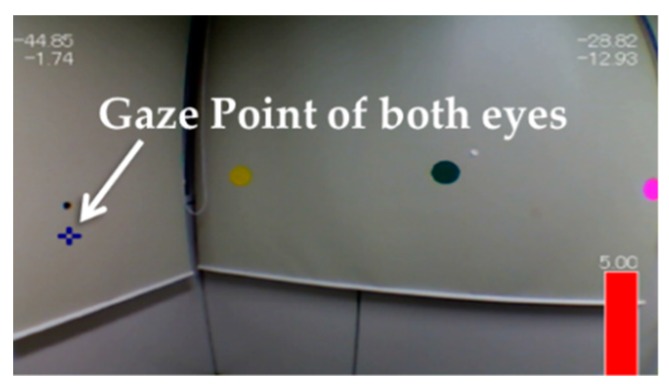
The operation image of using TalkEyeLite.

**Figure 3 sensors-19-04774-f003:**
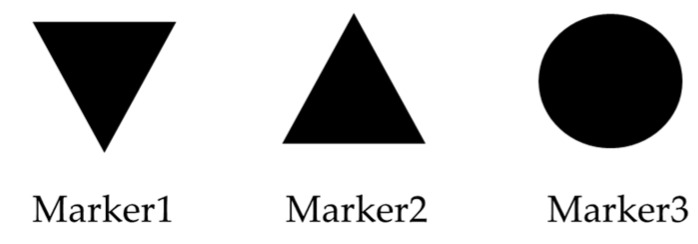
Types of markers.

**Figure 4 sensors-19-04774-f004:**
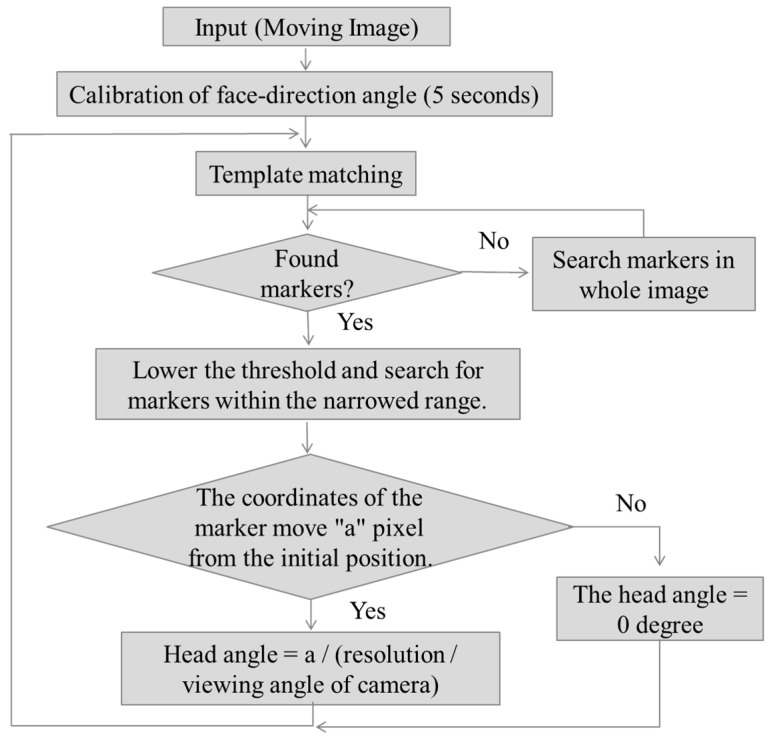
The flow of the head angle estimation method using template matching.

**Figure 5 sensors-19-04774-f005:**
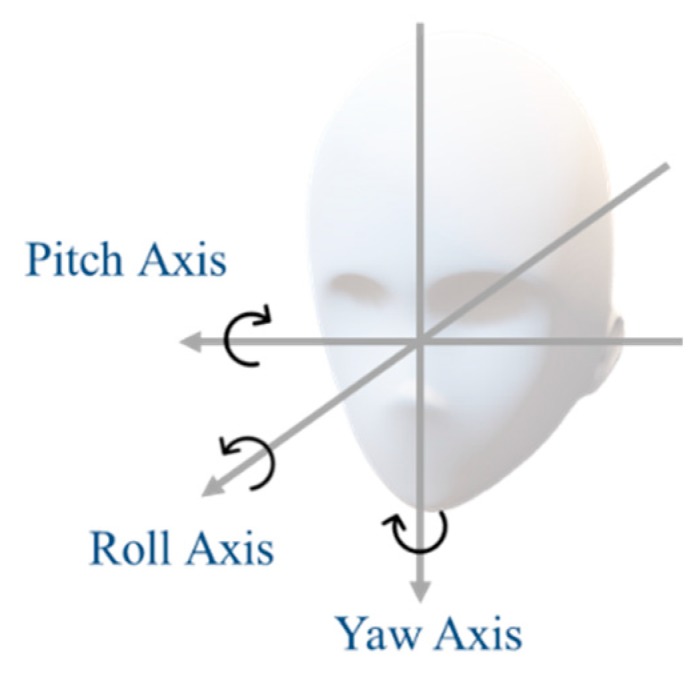
Head direction axes used for template matching.

**Figure 6 sensors-19-04774-f006:**
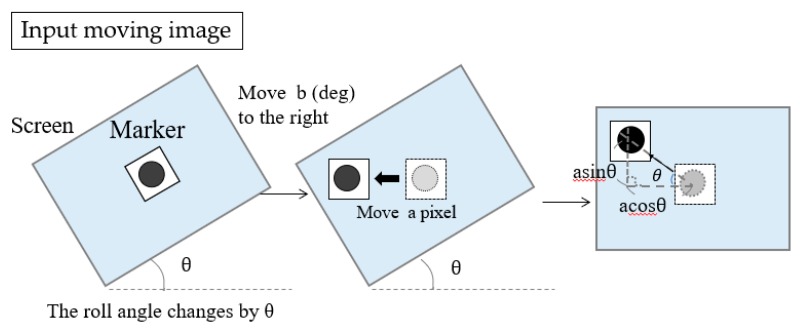
The flow of head angle estimation using template matching.

**Figure 7 sensors-19-04774-f007:**
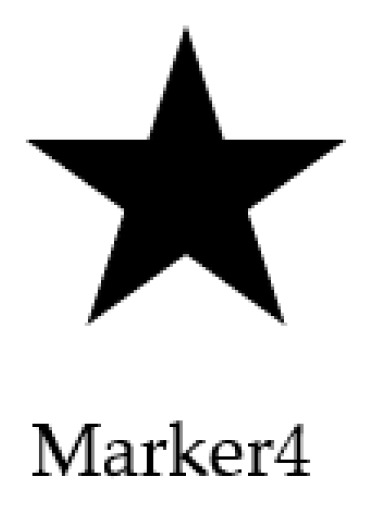
Newly added star marker (Marker4).

**Figure 8 sensors-19-04774-f008:**
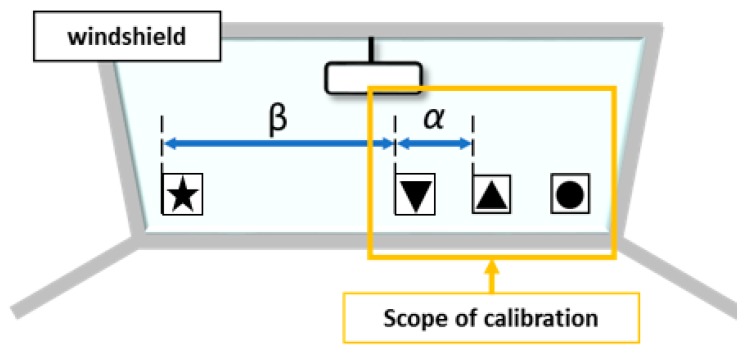
Between range to be calibrated and marker distance ratio. α, β means the ratio between markers.

**Figure 9 sensors-19-04774-f009:**
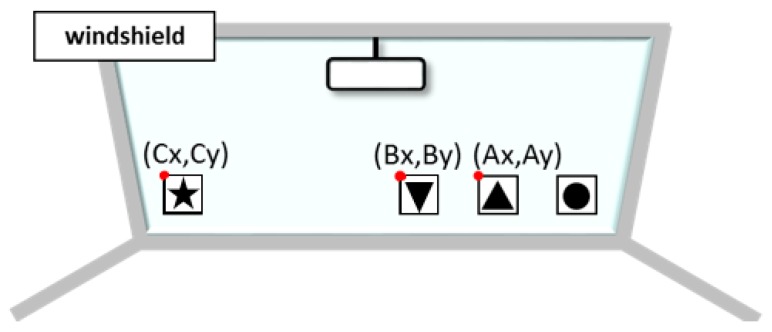
The coordinate positions of the markers.

**Figure 10 sensors-19-04774-f010:**
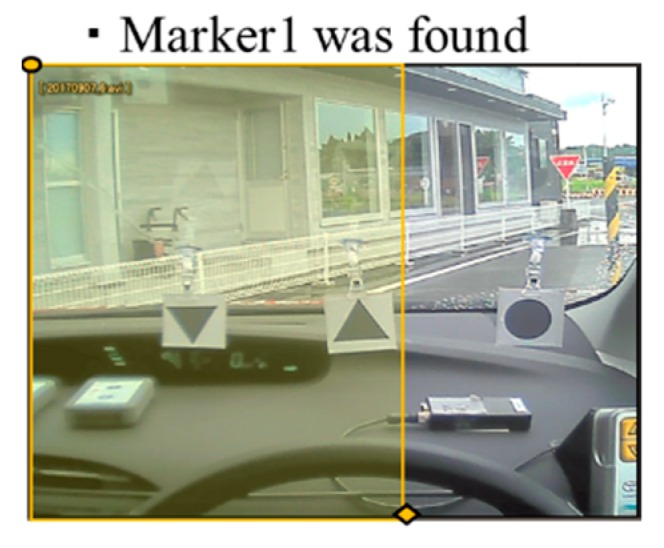
Search range for Marker2.

**Figure 11 sensors-19-04774-f011:**
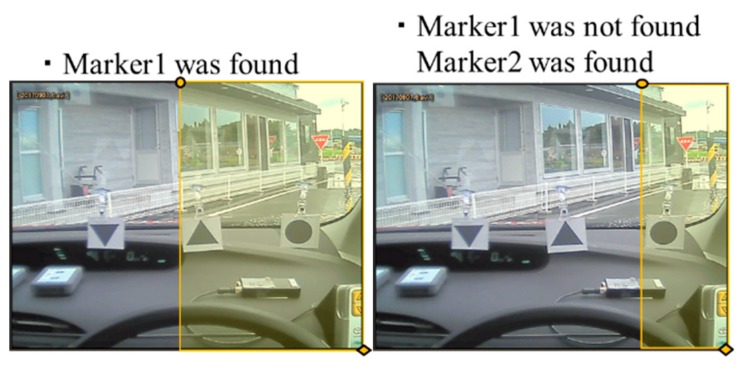
Search range for Marker3.

**Figure 12 sensors-19-04774-f012:**
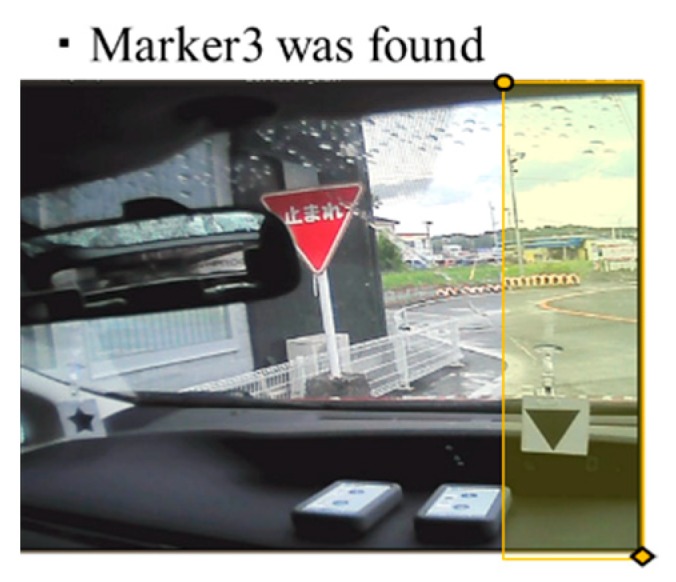
Search range for Marker4.

**Figure 13 sensors-19-04774-f013:**
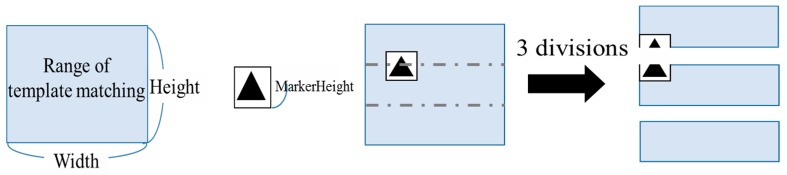
Marker problem.

**Figure 14 sensors-19-04774-f014:**
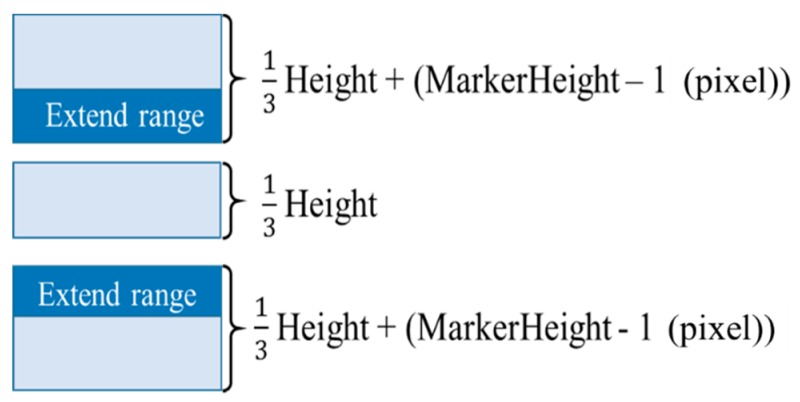
Improvement of the marker problem.

**Figure 15 sensors-19-04774-f015:**
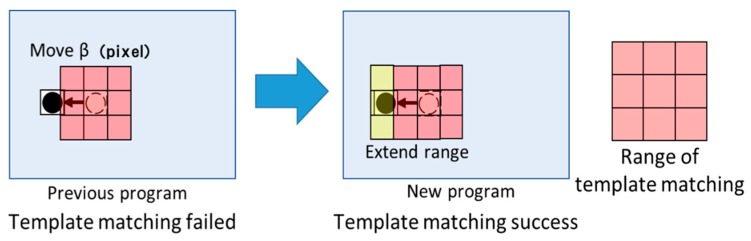
Example for extended search range.

**Figure 16 sensors-19-04774-f016:**
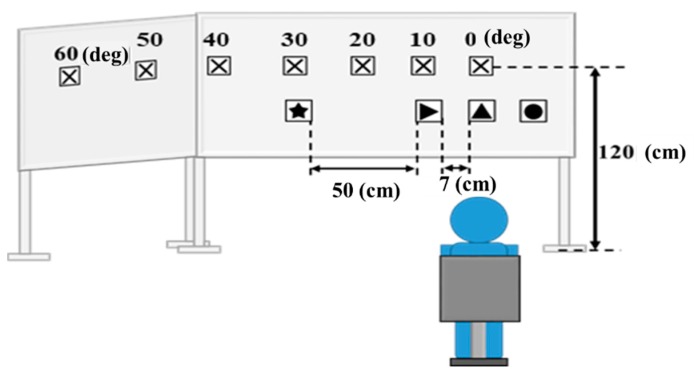
Experimental environment to verify the accuracy of head angle estimation using *Cx* coordinates.

**Figure 17 sensors-19-04774-f017:**
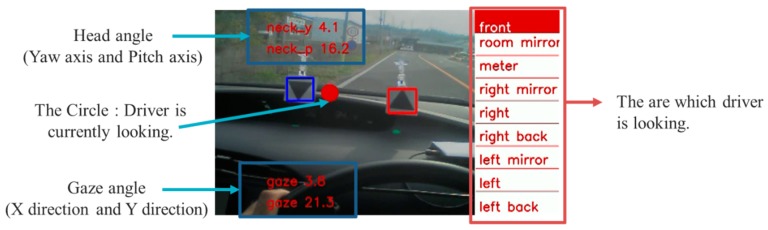
One of the results of our proposal system in an actual driving car.

**Figure 18 sensors-19-04774-f018:**
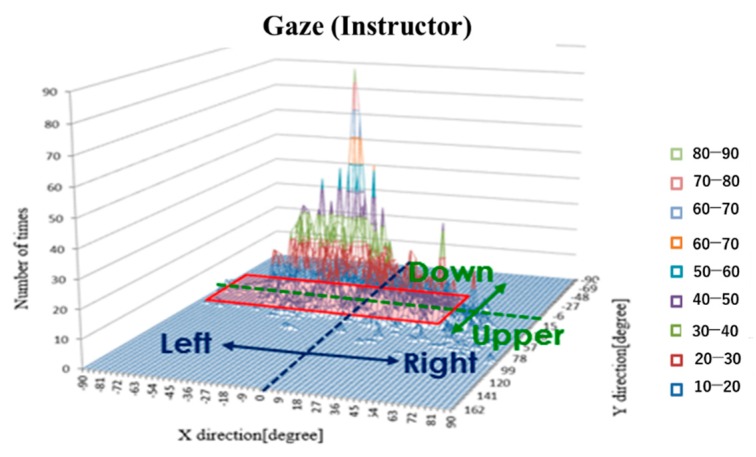
Example histogram of the gaze.

**Figure 19 sensors-19-04774-f019:**
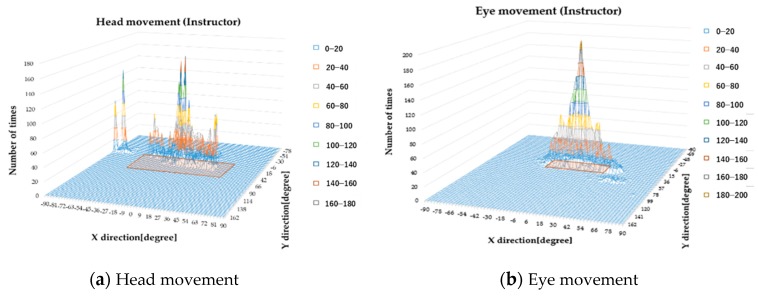
Histograms during car driving (Instructor).

**Figure 20 sensors-19-04774-f020:**
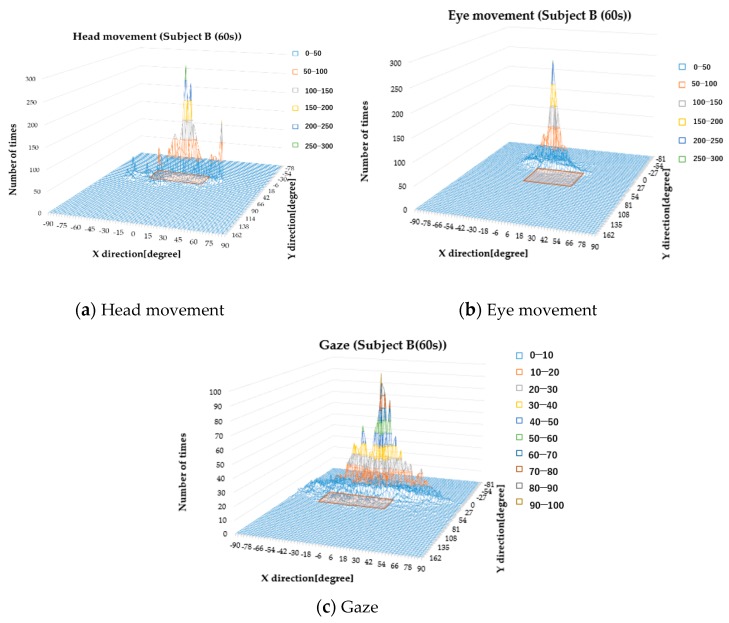
Histograms during car driving (subject B, 60–69 years old).

**Figure 21 sensors-19-04774-f021:**
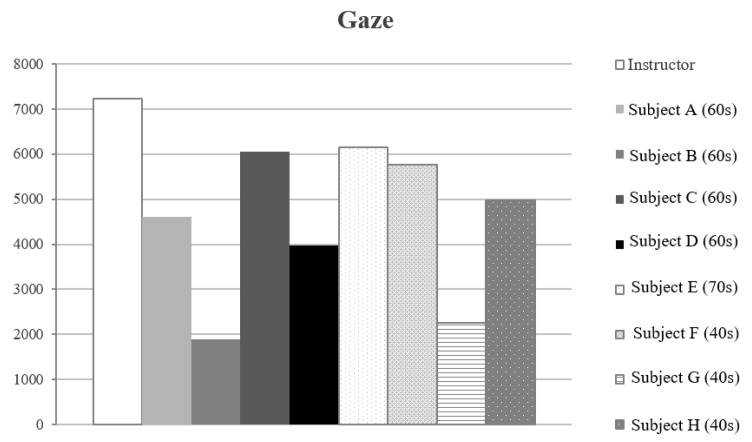
Compare the X × Y value.

**Table 1 sensors-19-04774-t001:** Average error of estimated head orientation angle.

True Value (deg)	Estimated Angle Error (deg)
10	1.9
20	4.3
30	6.1
40	7.3
50	4.1
60	0.9
Average	4.1

**Table 2 sensors-19-04774-t002:** Recognition accuracy of template matching.

Marker1 (Upward Triangle)	93.9%
Marker2 (Circle)	87.5%
Marker3 (Downward Triangle)	91.4%
Marker4 (Star)	88.2%

**Table 3 sensors-19-04774-t003:** Representative studies and mean error of head angle estimation.

Approach	Paper	Mean Error (deg)
Appearance Template Methods ■ Image Comparison	J. Ng and S. Gong [25]	9.6
Nonlinear Regression Methods ■ Neural Network	M. Voit et al. [26]	10.5
Manifold Embedding Method ■ Nonlinear Subspaces	S. Yan et al. [27]	7.8
Tracking Methods ■ Model Tracking	Y. Wu and K. Toyama [28]	24.5
Hybrid Methods	S. Ba and J.-M. Odobez [29]	9.4
Proposed Method ■ Template matching		4.1

**Table 4 sensors-19-04774-t004:** Comparison of marker recognition rate between before and after adding functions.

Marker	Before	After
Marker1 (Upward Triangle)	93.2%	93.7%
Marker2 (Circle)	85.1%	89.8%
Marker3 (Downward triangle)	90.8%	91.1%
Marker4 (Star)	83.7%	82.0%

**Table 5 sensors-19-04774-t005:** Comparison of processing time before and after adding functions.

Video	Before	After
Video1 (7 min 35 s)	15 min 1 s	10 min 48 s
Video2 (7 min 45 s)	15 min 20 s	10 min 59 s
Video3 (7 min 48 s)	16 min 1 s	11 min 33 s

**Table 6 sensors-19-04774-t006:** Results of outdoor experiments. The background colors mean that the actual situation and the Judgement flag are different.

Number	Situation	Time	Eye		Head		Gaze		Judgement Flag
			X_eye (deg)	Y_eye (deg)	X_head (deg)	Y_head (deg)	X_gaze (deg)	Y_gaze (deg)	
1	Left	01:28	−30.8	0.6	−25.3	−20.2	−56.1	−19.6	Left
2	Right	02:05	28.9	14.4	9.8	3.6	38.7	18	Right
3	Right	02:26	7.4	4.0	32.4	30.1	39.8	34.1	Right
4	Right	02:50	40.7	46.8	21.4	24.0	62.1	70.8	Right
5	Left	02:52	−11.5	9.0	−32.6	−18.8	−44.1	−9.8	Left
6	Left	03:13	−10.5	−3.4	−33.8	−16.2	−44.3	−19.6	Left
7	Left	03:22	3.4	2.1	−123.9	−12.4	−120.5	−10.3	Left
8	Left	03:38	−28.6	4.6	−112.4	−18.7	−141	−14.1	Left
9	Left	03:54	6.3	4.2	−96.6	−16.7	−90.3	−12.5	Left
10	Left	03:56	5.2	2.1	14.6	6.8	−29.5	16.8	Right
11	Left	04:27	−41.9	8.9	−35.0	−15.7	−76.9	−6.7	Left
12	Front	04:41	−5.4	6.8	33.9	11.5	−31.2	18.8	Left
13	Straight	04:45	−5.6	8.7	4.8	−20.4	−0.8	−11.6	Front
14	Left	05:26	8.7	3.7	−95.6	−21.9	−86.9	−18.2	Left
15	Left	05:50	−21.6	−0.5	−40.8	−17.8	−62.4	−18.3	Left
16	Left	05:51	−21.3	1.8	−40.8	−17.8	−62.1	−16	Left
17	Right	06:46	33.7	−20.8	−1.4	−11.2	32.3	−32.1	Right
18	Left	06:51	−24.6	0.4	−29.9	−13.8	−54.5	−13.3	Left
19	Right	06:56	22.8	−1.4	34.6	15.8	57.4	14.4	Right
20	Straight	07:03	2.0	−4.0	−9.3	−17.4	−7.4	−21.4	Front
21	Left	07:07	−5.9	−5.6	−121.4	−13.9	−127.3	−19.5	Left
22	Left	07:48	−19.8	−4.2	−22.1	−25.9	−41.9	−30.1	Left
23	Left	08:28	2.4	0.9	−119.7	−14.8	−117.3	−13.9	Left
24	Right	09:15	24.3	3.4	5.3	18.8	29.6	22.2	Right
25	Right	09:50	59.8	50.7	−3.1	−12.7	56.7	38	Right
26	Straight	10:13	−0.1	5.0	5.3	15.9	5.2	20.9	Front
27	Left	10:20	2.2	−1.9	−40.4	−7.7	−38.2	−9.5	Left
28	Right	10:23	17.7	8.8	25.0	14.7	42.6	23.4	Right
29	Straight	10:30	−5.1	−2.4	18.6	12.8	13.5	10.4	Front
30	Right	10:35	10.0	4.2	33.3	16.3	43.2	20.4	Right

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
