# Peer review of "A Study on the Gaze Range Calculation Method During an Actual Car Driving Using Eyeball Angle and Head Angle Information"

_sensors, 2019, doi:10.3390/s19214774_

Round 1

Reviewer 1 Report

The authors present a system that combines an off-the-shelf eye-tracker with a head tracking based on recognition of markers using OpenCV. They also propose a method to evaluate gaze range for car drivers and perform a small outside experiment with elderly drivers.

The work is interesting and has some merit. However, I feel this is not suitable for a publication at a journal. Authors should choose an appropriate conference venue to present their work in progress.

1° The method for head angle estimation using OpenCV detection of markers is interesting, but more thorough analysis and comparison with the state-of-the-art methods and commercially available eye-trackers is needed. Head tracking can be also performed remotely using a single RGB camera without wearing any devices (it should be quite uncomfortable for older drivers). Otherwise, low-cost precise solutions like Pupil Labs systems are available and can be used off-the-shelf.

2° The hypothesis of narrow gaze range implying more driving errors is extremely interesting but should be verified using an appropriate experimental design with more subjects and a proper statistical analysis (ANOVA or at least correlation analysis).

Author Response

Point 1: 1° The method for head angle estimation using OpenCV detection of markers is interesting, but more thorough analysis and comparison with the state-of-the-art methods and commercially available eye-trackers is needed. Head tracking can be also performed remotely using a single RGB camera without wearing any devices (it should be quite uncomfortable for older drivers). Otherwise, low-cost precise solutions like Pupil Labs systems are available and can be used off-the-shelf.

 Response 1: Thank you for your advice.

We have also performed head tracking using an RGB-D sensor. However, when the RGB-D sensor is used, there is a problem that cannot be estimated unless the head is moved within a narrow range. Also, when specializing in driving a car, installing the RGB-D sensor in front of the subject in a narrow space inside the car may hinder the securing of the field of view during driving. Therefore, this time, the angle of eye movement and head movement was estimated using a commercially available eye tracking device. The proposed method has the advantage that the gaze can be estimated with one device and the gaze can be estimated at a low cost.

Action: Added explanation on line 55 of page 2.

Point 2: 2° The hypothesis of narrow gaze range implying more driving errors is extremely interesting but should be verified using an appropriate experimental design with more subjects and a proper statistical analysis (ANOVA or at least correlation analysis).

Response 2: We also think that we need more subjects to analyze. As a result of performing statistical analysis (ANOVA) with the current number of subjects, we could not be said that age and gaze range were related. In the future, I would like to increase the number of subjects.

■ Action: Added explanation on line 494 of page 19.

We examined the gaze range and driving score at Discussion. However, since there were few subjects in this experiment, a significant statistical result was not obtained. Therefore, we deleted Discussion, and in this paper, we examined only the gaze range, which is an important item in car accidents in the elderly. Discussion is as flow.

■ Action: Deleted the Discussion.

Reviewer 2 Report

The authors are describing relationships between Gaze ranges and driving abilities.

I strongly suggest the authors hiring a native scientific writer. The paper on its current form lacks precision and direction. Almost every paragraph could be expressed in a more focused and concise manner.

On the bright side, the paper presents enough new information to be published, once properly and extensively edited. 

Author Response

The authors are describing relationships between Gaze ranges and driving abilities.

I strongly suggest the authors hiring a native scientific writer. The paper on its current form lacks precision and direction. Almost every paragraph could be expressed in a more focused and concise manner. On the bright side, the paper presents enough new information to be published, once properly and extensively edited.

Response: We examined the gaze range and driving score at Discussion. However, since there were few subjects in this experiment, a significant statistical result was not obtained. Therefore, we deleted Discussion, and in this paper, we examined only the gaze range, which is an important item in car accidents in the elderly. In the future, I would like to increase the number of subjects and examine the driving score and line-of-sight range again.

In addition, the native check was done.

 Action: Deleted the Discussion.

Reviewer 3 Report

The research regards interesting and important issues; however, there are some essential shortcomings in this work.

The description of the methods is insufficient:
─ Figure 6 should be better explained.
─ It was not explained why Marker4 was excluded from calibration.
─ Figure 8 includes two symbols ꞵ and α, which are not described.
─ Markers presented in figure 8 are not in the same order as in figure 3 and the order does not match to the relationships defined in lines 220-223.
─ Does the division presented in figure 13 take into account the division defined in figures 9, 10, and 11?
─ What does it mean ‘moved significantly right’ (line 266)? How was it measured?
─ Line 275, various positions for markers were pointed out. Does it mean that there were seven markers?
─ Is it sufficient to draw a conclusion based on an experiment involving only one person?
─ In lines 344--348, the gaze detection method was mentioned, but without any description. How was the eye angle calculated?
─ Line 396 - gaze range was mentioned - was it meant as a person’s ability or as a gaze range registered during experiments? How was this gaze range estimated?
─ Were the results presented in Table 6 obtained for one person or each participant?
─ Line 462 states that the description of the results regards participant E; however, all participants were addressed.
─ Figure 21 - what is the unit of gaze range?

Additionally, there is a lack of clear conclusions of the research. In my opinion, the number of participants was too small for valuable reasoning. Engaging young participants would allow checking the existence of differences (if exist) in gaze ranges for old and young drivers.

Minor comments:
Lines 302-312 present almost the same information as Table 3. Maybe, it could be simplified.

Author Response

Point 1: Figure 6 should be better explained.

Response 1: Thank you for your advice. We apologize for the lack of explanation.

If the yaw axis and the pitch axis change while the roll axis of the head is inclined, the head orientation angle cannot be estimated correctly. An example is shown in Fig.6. When the head roll axis (θ in Fig.6) is tilted, if the b [deg] moves to the right, the marker position moves a [pixel]. Therefore, when the roll axis rotates, we must first determine the roll angle. After calculating the roll angle, coordinate transformation was performed. The deviation of the estimated angle due to the movement of the roll angle is shown in Fig. 6. The deviation of the yaw axis can be calculated by ( a-acosθ ) and the deviation of the pitch axis by asinθ. By considering these deviations, we obtained coordinates converted to zero roll angle, solving the problem caused by inclination of the head.

 Action: Added explanation on line 147 of page 5.

Point 2: It was not explained why Marker4 was excluded from calibration.

Response 2: We apologize for the lack of explanation.Calibration is performed with the front facing. The marker 4 is excluded from calibration because it does not appear on the screen when facing the front. However, since it is necessary to estimate the face angle in a wide range, a reference position (Cx) of the marker was calculated, and a face angle estimation method including the marker 4 was proposed (chapter 4.1).

 Action: Added explanation on line 193 of page 6.

Point 3: Figure 8 includes two symbols ꞵ and α, which are not described

Response 3: We apologize for the lack of explanation. α: β means the ratio between markers.

 Action: Added explanation in Fig.8.

Point 4: Markers presented in figure 8 are not in the same order as in figure 3 and the order does not match to the relationships defined in lines 220-223.

Response 4: Corrected the marker order shown in Fig.3.

 Action: Revised the Fig.3.

Point 5: Does the division presented in figure 13 take into account the division defined in figures 9, 10, and 11?

Response 5: We apologize for the lack of explanation. Yes, we take into account the division defined in figures 9, 10, and 11.

Point 6: What does it mean ‘moved significantly right’ (line 266)? How was it measured?

Response 6: We apologize for the lack of explanation. Judge from the head movement angle and eye movement angle. We also judged from the visual field camera image.

 Action: Added explanation on line 272 of page 10.

Point 7: Line 275, various positions for markers were pointed out. Does it mean that there were seven markers?

Response 7: We apologize for the lack of explanation. In order to grasp the error of the head angle correctly, the indoor experiment was carried out assuming the optimum environment. The mark () indicates 0 to 60 degrees and is not a marker for template matching.

 Action: Added explanation on line 286 of page 10.

Point 8: Is it sufficient to draw a conclusion based on an experiment involving only one person?

Response 8: As you pointed out, it is not enough. This paper suggests that the proposed method may have a conclusion. In the future, I would like to repeat experiments not only for the elderly but also for patients with higher brain dysfunction.

Point 9: In lines 344--348, the gaze detection method was mentioned, but without any description. How was the eye angle calculated?

Response 9: The eye angle is calculated by TalkEyeLite. When the eyeball is irradiated with weak infrared light, a reflected image of the light source is generated on the refractive surface of the cornea or crystal. This reflection image, called the Purkinje-Sanson image, has the characteristic of little movement relative to the pupil during eye movement. Both are photographed with a camera, and the "eye movement angle" is calculated from the positional relationship.

 Action: Added explanation on lines 340 to 345 on page 12.

Point 10: Line 396 - gaze range was mentioned - was it meant as a person’s ability or as a gaze range registered during experiments? How was this gaze range estimated?

Response 10: The gaze range means a person’s ability. The eye movement angle was obtained by TalkEyeLite shown in Chapter 3, and the head movement angle was calculated using the template matching method. The gaze range was estimated using the value obtained by adding the head angle and the eyeball angle as the gaze.

 Action: Added explanation on lines 394 to 397 on page 15.

Point 11: Were the results presented in Table 6 obtained for one person or each participant?

Response 11: We apologize for the lack of explanation. Table 6 shows the results of an outdoor experiment with one elder driver.

Point 12: Line 462 states that the description of the results regards participant E; however, all participants were addressed.

Response 12: We apologize for the bad description. We defined X as Xmax-Xmin. Similarly, Y is Ymax-Ymin in the Y direction. The gaze range was calculated by X * Y.

Point 13: Additionally, there is a lack of clear conclusions of the research. In my opinion, the number of participants was too small for valuable reasoning. Engaging young participants would allow checking the existence of differences (if exist) in gaze ranges for old and young drivers.

Response 13: We think there are few subjects. As pointed out, I would like to include comparisons with young subjects in future experiments. Because the gaze range contributes significantly to car accidents, this paper evaluated driving ability by comparing the gaze range of elder people and driving instructors. We also proposed a method for enlarging the range of head movement angles using the template matching method and mentioned the accuracy of eye gaze estimation based on eye movement angles and head movement angles.

Point 14: Lines 302-312 present almost the same information as Table 3. Maybe, it could be simplified.

Response 14: Only the table was shown because the description and table contents overlapped.

We examined the gaze range and driving score at Discussion. However, since there were few subjects in this experiment, a significant statistical result was not obtained. Therefore, we deleted Discussion, and in this paper, we examined only the gaze range, which is an important item in car accidents in the elderly.

 Action: Deleted the Discussion.

Round 2

Reviewer 2 Report

The authors have satisfied my comments.

Reviewer 3 Report

Most of my comments were addressed. 

However, the concerns regarding the number of people engaged in evaluating the method and a lack of clear conclusions are still valid. 

This manuscript is a resubmission of an earlier submission. The following is a list of the peer review reports and author responses from that submission.

Round 1

Reviewer 1 Report

Most authors' names have a family name basis except several Japanese authors.

Author Response

Point 1: Most authors' names have a family name basis except several Japanese authors. 

Response 1: Thank you for your advice. We corrected the authors’ name because there were the errors in the method of describing the author's name in the reference (page 19 and 20).

Reviewer 2 Report

The final objective of the proposed research is to evaluate driving ability of older people. The eyeball angle and head angle estimation procedure is well described in the manuscript, but the estimated gaze range does not correspond with driving ability of older people. 

Author Response

Response to Reviewer 2 Comments

Point 1: The final objective of the proposed research is to evaluate driving ability of older people. The eyeball angle and head angle estimation procedure is well described in the manuscript, but the estimated gaze range does not correspond with driving ability of older people.

Response 1: Thank you for your advice. The advice of Reviewer2 is correct. We apologize for the lack of explanation. We will show you the way of thinking. We consider the following two points to be important in determining the driving ability of the older people.

Point 1. Automobile driving technology

For example, adjusting brake correctly, over speeding, oversight of signal etc.

Point 2. Range of vision during driving

Because it is known that the scope of vision becomes narrow as the aging phenomenon occurs in the older people. Therefore, when the field of vision is narrowed, traffic lights and pedestrians may be overlooked during driving. For this reason, we think that to estimate the range of vision during driving is important.

In this paper, point 1 was judged by the driving score in the driving school. And the point 2 used estimation of the visual field range at driving the car which this time proposed.

This paper only shows that there is a difference in the range of vision during driving between instructors and the older people. In the future, we will investigate the relationship between the driving score of the instructor and the older people and the gaze range and intend to create a system that enables comprehensive judgment of car driving ability from the relationship.

We added the missing description to the “Future Work” on page 18.

Round 2

Reviewer 2 Report

I recommend resubmitting after investigating the relationship between the driving score and the gaze range of the older people.